# Assessment of Drought and Zinc Stress Tolerance of Novel *Miscanthus* Hybrids and *Arundo donax* Clones Using Physiological, Biochemical, and Morphological Traits

**DOI:** 10.3390/biology12121525

**Published:** 2023-12-14

**Authors:** Monirul Islam, Andrea Ferrarini, Amjad Ali, Jason Kam, Luisa M. Trindade, John Clifton-Brown, Stefano Amaducci

**Affiliations:** 1Department of Sustainable Crop Production, Università Cattolica Del Sacro Cuore, Via Emilia Parmense 84, 29122 Piacenza, Italy; andrea.ferrarini@unicatt.it (A.F.); amjadparachinar@gmail.com (A.A.); stefano.amaducci@unicatt.it (S.A.); 2Department of Biochemistry and Molecular Biology, University of Massachusetts Amherst, Amherst, MA 01003, USA; 3Terravesta, Unit 4 Riverside Court, Skellingthorpe Road, Lincoln LN1 5AB, UK; 4Department of Plant Breeding, Wageningen University & Research, 6700 AJ Wageningen, The Netherlands; luisa.trindade@wur.nl; 5Institute of Biological, Environmental and Rural Sciences, Aberystwyth University, Aberystwyth SY23 3EB, UK; john.clifton-brown@agrar.uni-giessen.de; 6Institut für Pflanzenbau und Pflanzenzüchtung I, Justus-Liebig-Universität Gießen, Interdisziplinäres Forschungszentrum iFZ, Heinrich-Buff-Ring 26, 35392 Gießen, Germany

**Keywords:** bioenergy, *Miscanthus* hybrids, *Arundo* clones, drought tolerance, Zn tolerance, plant physiology, growth parameters, hierarchical clustering analysis (HCA), principal component analysis (PCA)

## Abstract

**Simple Summary:**

Marginal land is characterized by low crop productivity and is sometimes additionally contaminated. Such marginal land however presents a large opportunity to produce non-food biomass from perennial grasses with low risks of Indirect Land Use Change (low ILUC). *Miscanthus* spp. and *Arundo donax* also known as giant reed are leading bioenergy crops due to their high biomass productivity, but yields can be limited by insufficient water supply or phytotoxic levels of heavy metals. Drought and heavy metals are the most serious abiotic stress and negatively affect crop growth and development. The current study was conducted to identify the most drought and heavy metal (Zn) tolerant hybrid among seven novel *Miscanthus* hybrids and seven *Arundo* clones. Based on the morpho-physiological and biochemical analysis, the *M. sinensis* × *M. sacchariflorus* hybrid GRC 10 and *Arundo* clone PC1 were the most drought and Zn stress tolerant. The findings of this study provide a foundation for further investigations of the molecular and physiological mechanisms and recommendations for the cultivation of GRC 10 hybrid line and *Arundo* PC1 in marginal land.

**Abstract:**

High-yield potential perennial crops, such as *Miscanthus* spp. and *Arundo donax* are amongst the most promising sources of sustainable biomass for bioproducts and bioenergy. Although several studies assessed the agronomic performance of these species on diverse marginal lands, research to date on drought and zinc (Zn) resistance is scarce. Thus, the objective of this study was to investigate the drought and Zn stress tolerance of seven novel *Miscanthus* hybrids and seven *Arundo* clones originating from different parts of Italy. We subjected both species to severe drought (less than 30%), and Zn stress (400 mg/kg^−1^ of ZnSO_4_) separately, after one month of growth. All plants were harvested after 28 days of stress, and the relative drought and Zn stress tolerance were determined by using a set of morpho-physio-biochemical and biomass attributes in relation to stress tolerance indices (STI). Principal component analysis (PCA), hierarchical clustering analysis (HCA) and stress tolerance indices (STI) were performed for each morpho-physio-biochemical and biomass parameters and showed significant relative differences among the seven genotypes of both crops. Heatmaps of these indices showed how the different genotypes clustered into four groups. Considering PCA ranking value, *Miscanthus* hybrid GRC10 (8.11) and *Arundo* clone PC1 (11.34) had the highest-ranking value under both stresses indicating these hybrids and clones are the most tolerant to drought and Zn stress. In contrast, hybrid GRC3 (−3.33 lowest ranking value) and clone CT2 (−5.84) were found to be the most sensitive to both drought and Zn stress.

## 1. Introduction

The production of renewable energy from biomass crops has gained attention in European policies in recent decades for targeting the reduction of greenhouse gas emissions [1]. Particularly among the energy crops, perennial biomass crops became the core point of research interest due to their high potential yield and lignocellulose biomass quality [2,3,4,5,6]. Such perennial crops as *Arundo donax* also known as giant reed and genus *Miscanthus*, are traditionally used to produce bioenergy and bioethanol, bio-based products and anaerobic digestion [4,7,8,9,10]. To accomplish sustainable biomass for the bioeconomy, it is important to develop and identify genotypes that have better performance to abiotic stress tolerance and the ability to grow on underutilized marginal land to reduce the pressure on food production. In marginal lands, in particular, drought and heavy metals are two of the most limiting factors for crop production [11,12]. However, both *Arundo* and *Miscanthus* are suitable energy crop species in temperate marginal land due to their outstanding resilience and photosynthetic capacity at low temperatures [13]. Additionally, a prolonged drought in the summer months could limit the yields of these crops and threaten their survival [14]. On the other hand, heavy metal (HM) concentration in soil has rapidly increased because of various natural processes and anthropogenic (industrial) activities [15,16,17]. In Europe alone 137,000 km² of agricultural lands are contaminated with at least one or more heavy metals in higher concentrations than the threshold limit [18,19]. Among heavy metals, zinc (Zn) can be found in high concentrations in agricultural soils, which can damage cell functions, it can displace other elements having similar charges, such as Fe (iron), and Mg (magnesium) and reduce plant growth and increase chlorosis in leaves [20,21,22]. In the Earth’s crust, the average content of Zn is 70 mg kg^−1^ [23], and it varies from 10 to 100 mg kg^−1^ in soils around the world [24]. The availability of Zn for plant accumulation depends on its concentration in soil, the soil pH and soil clay fraction. Indeed, between 30 to 200 µg Zn g^−1^ dry mass (DM) is required as a micronutrient for most crop varieties to act in catalytic functions in several processes, like cell division, cell expansion, proteins, and carbohydrate metabolism [21,23]. However, at high concentrations in the soil (above 200 µg Zn g^−1^ dry mass (DM)), Zn toxicity inhibits water uptake and nitrate assimilation, which induces leaf water content, stomatal conductance, transpiration, net photosynthesis and photosynthesis efficiency [22,25]. The Zn toxicity threshold level widely depends on plant species, ranging from 100 to 500 mg (Zn) kg^−1^ (DM) [24]. Toxic levels of heavy metals combined with drought adversely affect plant physiology through several mechanisms, including photosynthesis, leaf water content, growth inhibition, and ROS (reactive oxygen species), which damage cells and modify membrane lipids [26,27,28]. This dual stress can be mitigated by several strategies, such as scavenging enzymes antioxidants, namely superoxide dismutase (SOD), peroxidase (POD), polyphenol oxidases (PPO), and ascorbate peroxidase (APX), which can reduce the negative impact of ROS [29,30]. Another survival strategy is to accumulate lower molecular weight organic solutes such as proline and phenols [31,32]. Long-term drought circumstances and high Zn negatively affected many physiological processes supporting biomass growth in *Miscanthus* spp. and *Arundo donax* (giant reed) species, according to several prior research and recent findings [24,33,34,35]. These perennial energy crops have exhibited high tolerance and restoration capacity to HMs stress by detoxification and accumulation mechanisms [24,36]. But to date, there is little known about the physiological and biochemical traits associated with drought and Zn stress tolerance among newly developed *Miscanthus* hybrids [37] and different clonal accessions of *Arundo donax* [4].

The main objectives of this study were to determine the drought and Zn stress conditions that (1) enabled discrimination between stress tolerant and susceptible *Miscanthus* hybrids and *Arundo* clones, as well as (2) to rank the responses using multiple traits of seven high-yielding *Miscanthus* hybrids and seven *Arundo* clones.

## 2. Materials and Methods

### 2.1. Plant Material, Growing Conditions and Experimental Design

The pot experiments were conducted from 2020 to 2022 in a growth chamber with a controlled environment at the laboratory of the Department of Sustainable Crop Production of the Università Cattolica del Sacro Cuore, Piacenza, Italy. All plant materials were collected from 4-years field trials in Piacenza (NW, Italy): rhizomes of *Miscanthus* hybrids were collected from a plot scale trials funded by the EU-BBI GRACE project [37] and clones of *Arundo* (Table 1) were collected from a self-funded field trials.

Rhizomes of *Miscanthus* for each hybrid from dormant mother plants were washed, cut into 7–10 cm lengths (around 10 g fresh weight) with several buds, and planted at a depth of 5 cm in 4 L circular plastic pots with a commercial blend of peat-humus, soil and sand (3:1:1). Propagation of *Arundo* clones by single-node stem cuttings was described earlier [38]. Briefly, the stems were cut and planted with the node at a depth of 1 cm below the surface in 4 L pots into the same compost described above. To increase the chances of *Arundo* stem survival, several stems per pot were planted and incubated in the dark for 10 days at a temperature of 25 °C (night 22 °C) and 55–60% humidity. After shoot emergence, a PPFD of 800 μmol m^−2^ s^−1^ was provided by light-emitting diodes (LED) in 16/8 h light/dark regimes for both species. Plants were watered to field water holding capacity (FWHC) every second day and fertilized weekly with a modified half-strength Hoagland’s solution (pH 6.0, EC 1.1 dS m^−1^). One month after germination, plants of both species were subjected to drought and Zn stress.

Experiments were carried out in a completely randomized block design (CRBD) with four biological replicates. The plants of seven hybrid lines and seven clones of both crops were well-watered to maintain the FWHC 60% as a control, for drought stress maintained 20% of FWHC (soil moisture maintained 1/3 of total FWHC), and for Zn stress once added 400 mg ZnSO_4_ × 7 H_2_O kg^–1^ of soil (dry mass, DM) in each pot. The pots were weighed every second day until the end of the experiment to maintain the desired water field water holding capacity (FWHC), control at 60%, drought at 20% and Zn stress at 60%. After 28 days from the onset of both stresses, all plants were tested for morphological measurements, physiological analysis (photosynthetic performance and relative water content (RWC) determination), and thereafter leaves were harvested in liquid nitrogen for further biochemical analysis. The harvested plant material was stored at −20 °C up to all analyses.

### 2.2. Growth and Biomass Characterization

Plant growth measurements were evaluated for all the hybrids and clones at the end of the treatment (28 days after treatment, DAT, and 58 days after sprouting). For each plant, plant height was measured by using a graduated ruler from the soil surface to the end of the ligule’s youngest fully expanded leaves. Similarly, the number of leaves was manually counted. At the end of the experiment, plants were harvested with leaves and stems (except the oldest expanded leaves for biochemical analysis) to determine aboveground shoots (with leaves) dry biomass. Biomass was dried at 60 °C in an oven for 72 h and subsequently, shoots of dry biomass were recorded.

### 2.3. Chlorophyll Fluorescence Measurements

Fluorescence measurements were taken on the last two fully expanded leaves of control and treated plants after 1-h dark adaptation. For each plant, measurements were performed with at least 4 technical replications (on a different portion of the same leaf) by using a Handy PEA Chlorophyll Fluorimeter (Hansatech Instruments Ltd., King’s Lynn, UK), with a one-second light pulse of 3500 μmol m^−2^ s^−1^ by three LEDs emitting at 650 nm. The initial and maximal fluorescence were determined to measure maximum photosystem II (PSII) photochemical efficiency Fv/Fm (ratio of variable fluorescence to maximum fluorescence). Additionally, the fast fluorescence transient [39] was measured for the determination of the performance index (PI).

### 2.4. Determination of Relative Water Content (RWC)%

The relative water content (RWC) of leaves with the same developmental stage was measured at harvesting. RWC was calculated using the following formula: RWC% = [(FW − DW/TW − DW)] × 100 (where; FW = fresh weight, DW = dry weight, TW = turgid weight). Fresh weight was taken immediately after cutting the leaves from plants, turgid weight after leaving the leaves 24 h in distilled water (in an aluminum tray at room temperature) and dry weight was measured after 72 h drying in an oven at 65 °C.

### 2.5. Biochemical Assays

#### 2.5.1. Analysis of Proline, Phenol and Malondialdehyde (MDA)

After 28 days of stress treatments (drought and Zn), the contents of proline and phenol were measured from leaf samples of both crops. In brief, free proline extraction was performed from frozen leaf tissues by grinding with a mortar and pestle [40]. Two hundred milligrams of ground samples were added to 5 mL of a 3% aqueous sulfosalicylic acid solution and vigorously vortexed for 1 min. Thereafter, the extract was centrifuged for 10 min at 4 °C at 10,000 rpm, and then the supernatant was stored at −20 °C in 2 mL Eppendorf tubes. For the determination of proline, 100 μL of the extraction was added with 1 mL of 1% ninhydrin solution which contains a 60:40 ratio of glacial acetic acid: water, and boiled at 95 °C in a water bath for 20 min. The reaction was stopped by submerging the samples in an ice bath. Thereafter, 3 mL of toluene was added and vigorously mixed by vortex and samples were left under dark conditions for 1 h. At the same time, the blank was prepared with 100 μL of a 3% sulfosalicylic acid instead of plant extraction. The light absorbance of the toluene phase was read at 520 nm with a microplate reader (Biotek Synergy 2, Winooski, VT, USA), and then proline concentration was determined by using a standard curve of proline. Results were expressed in μmol g^−1^ FW.

Total phenol contents in leaf tissues were determined through the Folin-Ciocalteu’s method [41]. One hundred milligrams of ground frozen plant material was added with 1.5 mL of 70% ethanol (*v*/*v*), vortexed, and centrifuged at 10,000 rpm for 5 min. Then 40 μL of the extract was mixed with 200 μL of Folin-Ciocalteu reagent and the solution was diluted by adding 1000 μL distilled water. Afterward, 600 µL of 20% sodium carbonate (Na_2_CO_3_, *w*/*v*) was added, samples were heated in a water bath at 85 °C for 1 min, then samples were allowed to stand at room temperature for 1 h in darkness. The absorbance of the samples read at 725 nm. The standard curve of gallic acid (GA) from the range 20, 40, 60, 80, 100, and 120 µg/mL was also prepared at the same time and finally, the results were expressed as total phenol contents mg GA g^−1^ FW.

The level of lipid peroxidation was quantified by measuring the production of malondialdehyde (MDA) in leaves after 28 days of stress treatments (drought and Zn) following the method described by [42]. 200 mg of ground fresh leave samples were mixed with 2 mL of reaction solution containing 0.5% (*v*/*v*) thiobarbituric acid (TBA) and 20% (*v*/*v*) trichloroacetic acid (TCA) and then samples were vortexed for 1 min. After incubation in a water bath at 95 °C for 30 min, the mixture was allowed to cool in an ice bath for 10 min, thereafter at room temperature, and then centrifuged at 10,000 rpm for 10 min. The absorbance of the supernatant was determined spectrophotometrically at 532 and 600 nm. The concentration of MDA was calculated using the formula: MDA (nmol g FW^−1^) = [(OD532 − OD600)]/(ε × FW), where FW is the fresh weight and ε the extinction coefficient (155 mM^−1^ cm^−1^). Data were expressed as μmol g FW^−1^ (fresh weight).

#### 2.5.2. Determination of Total Soluble Protein and Antioxidant Enzymes Activity

The total soluble protein was determined by using the kit of bicinchoninic acid (BCA) assay and standard of bovine serum albumin (BSA) (Thermo Fisher Scientific, Illinois, USA), as earlier described [43]. Briefly, plant material was ground with liquid nitrogen by mortar and pestle and added 0.1 M Na-phosphate buffer pH 7.0, containing 250 mM sucrose, 1 mM MgCl_2_, 1.0 mM EDTA, 0.1 mM dithiothreitol (DTT) and 1% (*w*/*v*) polyvinylpolypyrrolidone (PVPP) in a 1:10 proportion (plant material to buffer vol.). After this,the mixture was vortexed for 1 min before centrifugation at 12,000 rpm for 12 min. The supernatant was then used with BCA reagents in the development of intense purple color and read the absorbance at 562 nm with a microplate reader (Synergy HT Microplate Reader, BioTek Instruments, Inc., Winooski, VT, USA).

For the determination of POD (peroxidase) and PPO (polyphenol oxidases) enzyme activity, the method in ref. [44] method was followed with slight modification. PPO activity was measured as a catechol substrate, and the reaction was with 100 mM potassium phosphate buffer (pH 6.8), pyrogallol (50 µM) and 10 µL of enzyme solution in a volume of 200 µL. For POD activity the assay mixture contained 100 mM potassium phosphate buffer (pH 6.8), pyrogallol (50 µM), 10 µL of enzyme extract, and H_2_O_2_ (50 µM). For both activities, the absorbance was read at 420 nm and defined as an increase of 0.1 absorbance units. For the determination of SOD (superoxide dismutase) activity, 20 μL of plant extract was added with 0.1 mM EDTA, 50 mM NaHCO_3_ (pH 9.8), and at the end, 0.6 mM of epinephrine [45] and we waited four minutes to confirm the adrenochrome absorbance at 475 nm. To determine APX (ascorbate peroxidase) activity, 20 μL of plant extract was added with 50 mM potassium phosphate buffer (pH 7.0), 0.1 mM EDTA, 0.5 mM ascorbic acid, and lastly 0.1 mM H_2_O_2_. Afterward, the decrease in absorbance was measured from 30 s to 1 min at 290 nm, according to [46]. The APX activity was calculated based on the extinction coefficient (2.8 mM^−1^ cm^−1^).

### 2.6. Drought and Zn Tolerance Evaluation

To assess the drought and Zn stress tolerance of different genotypes, the stress tolerance index (STI) was used. STI was calculated using the following [47] formula:STI = (Yp × Ys)/(Ŷp)^2^, (1)
where Yp = value of each trait under control conditions, Ys = value of each trait under stress conditions, and Ŷp = mean value of all hybrids/clones under control conditions.

### 2.7. Statistical Analysis and Clustering

SPSS package (Version 26 SPSS Inc., Chicago, IL, USA) was used to analyze the data using a two-way-ANOVA (for genotypes, abiotic stress treatments, and their interactions) between control and treatments (drought and Zn) for all morphological (growth traits), physiological and biochemical analyses. Data were expressed as the mean ± standard deviation (S.D.). The significant differences between treatments mean were evaluated with Tukey’s HSD post hoc test at *p* < 0.05.

R statistical software (Version 4.1.1) was used for principal component analysis (PCA) and the ClustVis online tool (http://biit.cs.ut.ee/clustvis/, accessed on 10 September 2021) was used for Hierarchical clustering analysis HCA) based on STI values for each morphological (growth), physiological and physiological parameter. The drought and Zn stress tolerance of different hybrids of *Miscanthus* and clones of *Arundo* were assessed using PCA ranking value as earlier stated by [48] using the following formula:Ranking value = (Contribution of PC1 (%) × PC1) + (Contribution of PC2 (%) × PC2) + (Contribution of PC3 (%) × PC3). (2)

In this formula, two major components, PC1 and PC2, were obtained from PCA analysis and visually represented as percentages in the accompanying Figures. Both PC1 and PC2 are the PCA loading of morphological, physiological, and biochemical parameters for seven *Miscanthus* hybrids, and seven *Arundo* clones after 28 days of drought and Zn stress, separately. Finally, the numeric rank was calculated from the mean ranking values under drought and Zn treatments to evaluate and compare stress tolerance among the hybrids and clones.

## 3. Results

### 3.1. Growth Attributes and Biomass Accumulation

The effect of drought and Zn stress on the growth parameters and shoot dry biomass of seven hybrids of *Miscanthus* and *Arundo* clones are presented in Table 2. Based on the results, a significant reduction occurred in plant height and number of leaves per plant after 28 days of exposure to drought and Zn stress in the *Miscanthus* hybrids and *Arundo* clones (Table 2). At the same time, the effect of drought and Zn treatments and hybrids and clones, as well as the interactions between hybrids or clones and treatments, were significant (*p* < 0.05) for all morphological and physiological and biochemical parameters, except shoot dry weight (SDW) and photosynthesis performance index (PI) traits of *Miscanthus* hybrids and *Arundo* clones, respectively. The *Miscanthus* hybrid GRC10, and *Arundo* clone PC1 showed higher plant height than the other six hybrids and clones, under drought and Zn stress conditions compared with control plants. On the other hand, the most significant reduction in plant height and number of leaves in both treatments was observed in *Miscanthus* hybrids GRC3 and GRC6, and *Arundo* clones CT2 and PI1.

Under well-watered (control) conditions, *Miscanthus* GRC10 followed by GRC14 and *Arundo* PC1, followed by PC7 and PC6, showed the highest levels of shoot-dry biomass accumulation. The highest decrease of shoot-dry weight (SDW) biomass was observed in *Miscanthus* GRC6, GRC9 (54.0 and 54.5%) and GRC3 (48.2%), and *Arundo* CT2 (54%), PI1 (53%) under drought stress while a similar trend of decreasing was observed under Zn stress. The smallest decrease in SDW, in comparison to the control plants was measured on the hybrid GRC10 (8% drought and 4% Zn) and *Arundo* clone PC1 (19% drought and 16% Zn) under drought and Zn stress.

### 3.2. Physiological Responses to Drought and Zn Stress

#### 3.2.1. Chlorophyll Fluorescence under Drought and Zn Stress

The dark-adapted maximum quantum yield of PSII, as Fv/Fm and performance index (PI) sharply declined under drought and Zn toxicity. The decline of PSII (Fv/Fm), PI and their significant interactions under treatments compared to control conditions for all hybrids and clones are shown in Table 3. GRC6, and GRC3 *Miscanthus* hybrids showed the highest reduction in Fv/Fm (62 and 52%) while *Arundo* PI1 and PC7 clones showed 38 and 31% respectively, under drought conditions. A less significant decline was observed in *Miscanthus* GRC10 (5% decrease) while no significant difference was measured in *Arundo* clone PC1 (2% decrease) in drought stress compared with control. On the other hand, under Zn stress, *Miscanthus* GRC9 (62.5%), GRC15 (60.7%) and GRC3 (52%), and *Arundo* PI1 (50%) showed the highest reduction in Fv/Fm, whereas the maximum quantum efficiency of hybrids GRC10 and PC1, PC6 clones were maintained under such condition.

The PI was significantly reduced for all hybrids of *Miscanthus* after 28 days of stress (drought and Zn), while no significant difference was observed in *Arundo* PC1, PC6, and PC7 clones (Table 3). Among the *Miscanthus* hybrids, GRC10 decreased less (9 and 10%) compared with the more sensitive GRC3 (63 and 52%), and GRC6 (68 and 48%) under both drought and Zn stress, respectively. *Arundo* clone PC1 showed a higher PI compared to all clones under both stress conditions (Table 3). The highest reduction of PI occurred in *Arundo* CT2 (42.6 and 34.4%, drought and Zn) and PI1 (22.7 and 31%, drought and Zn) clones under drought and Zn stress (Table 3). Table 4 represented all morpho-physio-biochemical and biomass attributes interactions in treatments (Zn and drought), hybrids and clones.

#### 3.2.2. Effect of Drought and Zn Stress on Water Content (RWC)

Relative water content (RWC) was measured to assess plant water status either under control or stress conditions. All *Miscanthus* hybrids and *Arundo* clones showed a high level of leaf RWC (*Miscanthus* values between 71 to 95% and *Arundo* 87 to 93%, respectively) under control (well-watered) conditions. Significant effects (*p* < 0.05) on RWC were observed for all hybrids and clones under drought and Zn stress conditions in comparison to control plants. The highest reduction of RWC occurred in *Miscanthus* GRC3 in both drought and Zn stress (25 and 23%, drought and Zn) and GRC1 (11 and 22%) and under Zn stressed GRC15 (22.4%). *Miscanthus* GRC10 showed a lower reduction of RWC under drought and Zn stress (8 and 6%, drought and Zn) than control plants. On the other hand, in *Arundo*, most of the clones displayed a reduction of RWC under drought and Zn stress. The highest RWC reduction was observed in *Arundo* CT2, PI1 (drought 28 and 29% and Zn 33 and 23%, respectively) and moderate reduction was recorded in clones A1, PC6, and PC7 (between 16 to 23%) under both stresses.

### 3.3. Biochemical Responses to Drought and Zn Stress

#### 3.3.1. Effect of Drought and Zn Stress on Proline and Phenol Accumulation

Proline content (PC) and total phenol content (TPC) were significantly affected under drought and Zn stress (Figure 1A–D). In *Miscanthus* GRC10 and *Arundo* PC1 showed the highest PC and TPC, while the smallest PC and TPC contents in *Miscanthus* GRC3, *Arundo* ASR, and CT2, respectively (Figure 1A,B). PC content considerably increased under drought and Zn stress conditions in all *Miscanthus* hybrids (except GRC3) by a fold increase ratio from 2.22 to 3.36, similarly, all clones of *Arundo* (except CT2) also increased by a fold change ratio from 1.44 to 2.54 compared with the control condition (Figure 1A,B). A similar trend in TPC accumulation was observed in all *Miscanthus* hybrids by fold change ratio from 1.47 to 2.17 while in *Arundo* clones PC1, PI1, ASR, and PC6, there was a fold change ratio of 1.16 to 1.58 under both stresses compared with control groups (Figure 1C,D). However, the highest accumulation of both TPC and PC was measured in *Miscanthus* GRC10 (TPC fold-change ratio 2.17, 1.98; PC ratio 2.45, 2.75 droughts and Zn, respectively) (Figure 1A,C) and in *Arundo* PC1 (TPC fold-change ratio 1.41, 1.37; PC ratio 2.54, 2.42 drought and Zn, respectively) under both drought and Zn stress (Figure 1B,D).

#### 3.3.2. Lipid Peroxidation under Drought and Zn Stress

Malondialdehyde (MDA) content is an important indicator regarding plant oxidative stress and redox signaling, and long exposure to drought or Zn stress in certain *Miscanthus* hybrids and *Arundo* clones increased MDA in leaves (Figure 2). Particularly, compared with the control conditions, the highest increase in MDA was recorded in the *Miscanthus* hybrids GRC3 (229 and 378%, drought and Zn, respectively), GRC6 (64 and 67%) (Figure 2, and in the *Arundo* clones PI1 (292 and 170%), PC7 (329% and 62%) and A1 (216 and 174%) under drought and Zn stress, respectively (Figure 2). A significant increase in MDA occurred under Zn stress in hybrid GRC14 (111%) and *Arundo* ASR (184%). Under such conditions (drought or Zn stress) the lowest accumulation was recorded for *Miscanthus* GRC10 (16 and 25%, drought and Zn, respectively) and *Arundo* PC1 (22 and 15%, drought and Zn, respectively).

#### 3.3.3. Effect of Drought and Zn Stress on Soluble Protein Accumulation and Activities of Antioxidant Enzymes

Soluble protein is an essential component for cellular osmotic regulations. The total content of soluble protein increased with the prolongation of drought and Zn stress in *Miscanthus* hybrids (except GRC9) and *Arundo* clones (except PC7), as reported in Figure 3. Under drought stress compared to control conditions, the total soluble protein content highly increased in *Miscanthus* GRC1 (70%), GRC3 (70%) and GRC10 (20%) (Figure 3A), and *Arundo* A1 (82%), PC1 (20%) and PC7 (15%) (Figure 3), but significant reduction occurred in GRC9 (18%) and PC7 (15%), respectively. On the contrary, the accumulation of total soluble protein showed a similar trend under Zn stress for *Miscanthus* GRC1, GRC3 and GRC10 hybrids and *Arundo* A1, PC1 and PC7 clones (Figure 3).

The activities of all antioxidant enzymes including PPO, POD, SOD and APX in response to drought and Zn stress on *Miscanthus* hybrids and *Arundo* clones are shown in Figure 4A–H. However, the increasing and decreasing effect of antioxidant enzymes depended on both *Miscanthus* hybrids and *Arundo* clones. The analysis of PPO activity showed a gradual increase in leaves of *Miscanthus* GRC10, GRC9 and GRC1 and all the clones of *Arundo* under both drought and Zn stress (Figure 4A,B). Compared with the control condition, under drought and Zn stress, PPO activity revealed the highest increase in GRC10 (32 and 48%, drought and Zn stress, respectively), GRC9 (31 and 45%) (Figure 4A) and in *Arundo* PC1 (37 and 42%), PC7 (30 and 40%) and PI1 (27% in both stresses) (Figure 4B). On the other hand, such enzyme activity in *Miscanthus* GRC3, GRC14 and GRC15 and *Arundo* ASR, CT2, and PC6 remained unchanged under both drought and Zn stress (Figure 4A,B). In the case of POD, *Miscanthus* GRC9 (35 and 40%, drought and Zn, respectively), GRC10 (23 and 15%) and *Arundo* PC1 (103 and 52%) and PC7 (63 and 61%) showed the highest activity under such drought and Zn stress (Figure 4C,D). In contrast, *Miscanthus* GRC1 under drought stress and *Arundo* A1 under both drought and Zn stress showed a significant *(p < 0.05)* reduction in POD activity. *Miscanthus* GRC3, GRC14 and GRC15 and *Arundo* ASR, CT2 and PC6 remained unchanged for such enzyme activity (Figure 4C,D).

SOD activity increased in all *Miscanthus* hybrids, except GRC1, GRC15, and all giant clones due to drought and Zn stress (Figure 4E,F). The maximum increase in SOD activity was found in *Miscanthus* GRC10 (30 and 38% drought and Zn stress, respectively), GRC9 (21 and 35%) and GRC14 (11 and 25%), followed by *Arundo* A1 (196 and 191%), PC1 (75 and 97%) and ASR (32 and 20%) under drought and Zn treatment, respectively. Regarding the SOD activity, *Miscanthus* GRC3, GR6 and GRC15 remain unchanged like POD in response to drought and Zn stress (Figure 4E,F).

In the analysis of APX enzyme activity, all *Miscanthus* hybrids, except GRC3 and all *Arundo* clones, except PC6 and PC7 showed higher peak increase under drought and Zn stress, and among *Arundo* PC7 decreased significantly under such stresses (Figure 4G,H). Compared with the control, APX significantly increased under drought stress in *Miscanthus* GRC10, GR6 and GRC14 and *Arundo* A1 and PC1 by 40%, 31%, 28%, 98% and 41%, respectively. A similar increasing trend of APX activity was found for these hybrids and clones under Zn stress compared with control. However, APX activity was unchanged in *Miscanthus* GRC3 and followed by GRC9 and *Arundo* ASR and PI1 (Figure 4G,H).

### 3.4. Ranking among the Hybrids and Clones to Drought and Zn Stress Tolerance

The loading plots of principal components 1 and 2 for the seven *Miscanthus* hybrids and seven *Arundo* clones under drought and zinc stress conditions are displayed in Figure 5A–D. These plots are based on an examination of the growth, physiological, and biochemical characteristics. The results from principal components analysis (PCA) of *Miscanthus* hybrids under drought and Zn stress showed that principal component 1 (PC1) explained approximately 53.4% and 61.2% of the total variations (Figure 5A,B), and the second principal (PC2) 18.2% and 16.2%, respectively (Figure 5A,B) (Appendix A, Appendix A). On the other hand, concerning PCA of *Arundo* under drought stress and Zn treatment, PC1 explained 62.1% and 61.1% (Figure 5C,D) of the total variance, while the second principal component (PC2) explained 15.7% and 16.7%, respectively (Figure 5C,D) (Appendix A, Appendix A). Regarding *Miscanthus* hybrids under drought stress, the first component (PC1) was characterized by a high positive score with SDW, phenol, PI, PH and proline (Figure 5A) and under Zn stress, PPO, SDW, PI, Fv/Fm, and APX which were negatively correlated with MDA (Figure 5B). Under drought stress in *Miscanthus* hybrids, the second component (PC2) was identified with a high score with RWC and APX, while for Zn stress, it was phenol and RWC (Figure 5B). On the other hand, in *Arundo* clones, under drought stress PC1 was characterized by high positive score proline, NOL, POD, PPO, SDW phenol, and protein which were negatively correlated with MDA, and under Zn stress were PPO, NOL, SDW, proline and POD (Figure 5C). However, under Zn stress, MDA showed a negative correlation with SOD and a positively correlated with Fv/Fm (Figure 5D) in *Arundo* clones. The second component (PC2) in *Arundo* clones was identified with high score protein and SOD which were negatively correlated with MDA. Therefore, the *Miscanthus* hybrids and *Arundo* clones with high PC1 and PC2 scores had better performance in either drought or Zn stress compared to other hybrids or clones.

The heatmap from hierarchical clustering analysis (HCA) showed morphological, physiological, and biochemical parameters under drought and Zn stress could be clustered into four distinct groups (Figure 6A–D). According to the color scale, the dark red color represents the highest values, while the yellow represents the lower STI values of the parameters under drought and Zn treatments. In agreement with PCA observations, 14parameters were grouped into different clusters under drought and Zn stress of *Miscanthus* hybrids and *Arundo* clones (Figure 6A–D). The heatmap categorized the seven hybrids and seven clones into four distinct clusters A, B, C, and D based on the results of STI from drought and Zn treatment (Figure 6A–D). Cluster A indicated GRC10 was the most drought and Zn tolerant (highlighted by dark red color), and cluster B including GRC14 and GRC15 indicated moderate drought-tolerant hybrids (Figure 6A,B). *Miscanthus* GRC1, GRC3, GRC6 and GRC9 indicated the most sensitive hybrids of both drought and Zn stress in clusters C and D, respectively (Figure 6A,B). On the other hand, *Arundo* clones were also indicated in four clusters, whereas cluster A including PC1 and then ASR indicated the most drought tolerant (Figure 6C) while PC1 was only the most Zn stress-tolerant. For cluster B, PC6 and A1 indicated moderate drought and Zn stress-tolerant (Figure 6C,D). The clones, including CT2 and PI1, were clustered in group C, which indicated the most sensitive clones for drought and Zn stress tolerance (Figure 6C,D).

In addition to the PCA and HCA results, a mean ranking value was calculated to represent drought and Zn tolerance for *Miscanthus* hybrids and *Arundo* clones. *Miscanthus* GRC10 then GRC14 and GRC15 displayed higher mean ranking values under drought stress whereas under Zn stress GRC10 and GRC 14 showed higher mean ranking values (Table 5). In contrast, *Miscanthus* GRC3 and GRC9 showed comparatively lower mean ranking values under both drought and Zn stress, suggesting these hybrids were more sensitive to drought and Zn stress. Additionally, among the *Arundo* clones, PC1 and ASR showed higher mean ranking values compared to other clones under drought stress, while under Zn stress PC1 and PC6 demonstrated higher mean ranking values (Table 5). *Arundo* CT2 and PC7 revealed lower mean ranking values among the seven clones, suggesting that these two clones were more sensitive to both drought and Zn stress (Table 5).

## 4. Discussion

The plant exhibits several responses to drought and HM stress conditions, and one of the most critical responses is reducing the growth rate. However, plant development and biomass production under drought or toxic metal conditions are associated with the better adjustment of water relations to sustain physiological and biochemical activities [34,49]. In the current study, all *Miscanthus* hybrids and *Arundo* clones exhibited severe responses under both drought and Zn stress treatments.

In bioenergy crops, biomass yield is a key factor for determining economic viability [50]. In previous studies, growth, biomass, and physiological response under drought stress [34,51,52], and under Zn stress [24,36] in *Miscanthus* spp. were hybrid-specific, while in *Arundo* were clone or ecotype-specific [36,53,54]. Both drought and Zn stress negatively affected the growth of the *Miscanthus* hybrids and *Arundo* clones in the present study. In agreement with the previous studies, the growth response was hybrid-specific in *Miscanthu*s and clone-specific in *Arundo* under both drought and Zn stress. The decrease in total dry weight including stem and number of leaves per shoot under drought or Zn stress might be related to accelerated leaf senescence, reduced stomatal conductance, decreased photosynthesis and increased suppression of cell growth under low turgor pressure [52,55,56].

Chlorophyll fluorescence is known as an effective technique to monitor the physiological status of plants under several abiotic stress [57]. The maximum quantum (Fv/Fm) of photosystem II and performance index (PI) are efficient parameter for discriminating among tolerant or sensitive *Miscanthus* hybrids and *Arundo* clones. In contrast, *Miscanthus* GRC10 andGCR15 and *Arundo* PC1 and ASR showed higher Fv/Fm and PI under both drought and Zn stress. Similar findings were reported in several edible crops and non-food crops tolerant varieties under such stress [34,52,58].

The major consequence of drought is related to water status in plants which defines a plant’s potentiality to survive under the water-deficient condition to maintain growth, and photosynthesis activity. Meanwhile, heavy metals limit the water uptake [24]. As stated earlier by [59], relative water content (RWC) is one of the common indexes for assessing plant water status. However, increased water retention during dehydration is a crucial static for developing drought resistance [60]. Generally, it has been shown that exposure to drought stress changed RWC in several crop species including wheat [61,62], tomato [63], and sorghum [64], for both drought and Zn stress in *Miscanthus* spp. [24,34]. Previous studies confirmed the relationship between RWC and plant genotypes’ ability to tolerate drought, with the maintenance of a relatively constant RWC widely regarded as one of the best criteria for identifying and selecting tolerant and sensitive genotypes [34,63]. In this study, a high RWC was recorded in *Miscanthus* hybrids of *M. sinensis* × *M. sacchariflorus* GRC10 and then GRC14, while *Arundo* PC1, PC6 and ASR clones that were the most tolerant hybrids and clones to drought and Zn stress in our study.

Plants have evolved a wide array of different pathways to respond to different stresses, specifically through secondary metabolites production [65,66]. Among these, phenolic compounds are important secondary metabolites, whose accumulation in plants increases abiotic stress tolerance [67]. Our results revealed that most of the *Miscanthus* hybrids and *Arundo* clones increase total phenol contents (TPC) under both drought and Zn stress, and such accumulation was highest in hybrids *M. sinensis* × *M. sacchariflorus* GRC10, then GRC14, and GRC15 and in *Arundo* clones PC1, ASR and PC6. The highest accumulation was observed in *Miscanthus* spp. under drought conditions [68] in *Arundo* under nickel (Ni) and copper (Cu) stress [49]. Malčovská et al. [69] proposed that plants increase TPC in cells when plants are exposed to heavy metal stress as phenol are reactive oxygen species scavengers along with metal chelators. On the other hand, soluble protein and proline are considered compatible solutes and osmoregulators and they play an adaptative function in stress tolerance in higher plants [70]. In our results, soluble protein contents increased highly in *Miscanthus* hybrids of *M. sinensis* × *M. sacchariflorus* mostly GRC10 and GRC14 and *Arundo* clones PC1 and ASR under both drought and Zn stress. Similar results were found in rice drought stress-tolerant varieties [71] and in *Miscanthus* cadmium (Cd) stress-tolerant species [72]. Proline has been shown to be an energy supplier in membrane and subcellular structures, protect the plant’s photosynthetic apparatus by a radical oxygen scavenger and maintain the redox potential that enables plant’s growth, development, and survival under stress conditions [60,73,74]. The level of accumulation of proline under stress conditions was used to identify stress-tolerant genotypes, as reported by several studies [60,73], of *Miscanthus* species [72]. However, contrary results have also been reported in a correlation between the degree of stress tolerance and proline accumulation [73,75]. In our study, *Miscanthus* hybrids of *M. sinensis* × *M. sacchariflorus* GRC10 and thereafter in GRC15 and *Arundo* clone PC1 showed the highest accumulation of proline under both drought and Zn stress that were the most tolerant hybrids and clones. Considering that the highest dry biomass (Table 2) was also observed in the hybrids of *M. sinensis* × *M. sacchariflorus* largely GRC10, and then GRC15 and *Arundo* PC1 and A1 clones, this might indicate that proline, being involved in cytoplasmic osmotic adjustment, could enhance drought and Zn stress tolerance in *Miscanthus* GRC10, and GRC15 and *Arundo* PC1 and A1 clones.

Drought and heavy metals, like other abiotic stresses, increase the formation of reactive oxygen species (ROS) that damage plants. As a general adaptation strategy, plants utilize a few enzymatic and non-enzymatic detoxification systems to protect themselves from oxidative damage caused by ROS [63,76]. Among the antioxidant’s enzymes, it includes CAT, POD, PPO, SOD, APX and non-enzymatic systems, such as phenol and proline which works together to support plants to survive under stress conditions [77]. Nevertheless, such antioxidant enzyme mechanisms are complicated and genetically controlled [78]. In our study, *Miscanthus* hybrids of *M. sinensis* × *M. sacchariflorus*, specifically GRC10 and then GRC14 and GRC15, and *Arundo* clones PC1, ASR and PC6 showed higher activity of PPO, SOD and APX enzymes under both drought and Zn stress which is linked with the lower accumulation of MDA and higher accumulation of proline, phenol and better photosynthesis performance. SOD and POD could play a great role in catalyzing H_2_O_2_ into H_2_O and oxygen produced by ROS H_2_O_2_, meanwhile PPO produces rich phenolic compounds [71]. In agreement with past studies, *Miscanthus* tolerant species highly increased PPO, SOD, and APX activity under drought [36] and heavy metal (Cd, Ni, and Zn) stress [33] and *Arundo* drought and Cd stress [79]. Thus, our results support that drought and Zn stress tolerance capability among the different hybrids and clones are positively correlated with PPO, SOD and APX antioxidants activity.

In the present study, PCA and HCA were combined used to cluster seven hybrids of *Miscanthus* and seven *Arundo* clones with varying drought and Zn tolerance into four major groups based on their growth and physiological parameters. PCA analysis showed that the variation among the seven hybrids of *Miscanthus* and seven *Arundo* clones was largely due to their alternations in growth and physiological parameters (Figure 5A–D). According to the heatmaps, clusters A and B showed better growth, lower MDA, and a higher accumulation of proline under both drought and Zn stress. These clusters had higher mean ranking values among the *Miscanthus* hybrids and *Arundo* clones which comprised *Miscanthus* GRC10 and GRC15 hybrids and *Arundo* PC1 and ASR clones, suggesting these hybrids and clones have good tolerance to drought and Zn stress. However, the most drought and Zn-tolerant *Miscanthus* hybrid GRC10 and *Arundo* clone PC1 were placed in cluster A, especially due to relatively higher levels of proline, phenol and higher growth and lower MDA. Cluster C and D indicated lower growth and lower proline, phenol and higher MDA under both drought and Zn stress for *Miscanthus* hybrids and *Arundo* clones. Based on HCA data, *Miscanthus* GRC3 and *Arundo* CT2 found the most drought and Zn- sensitive hybrids and clones which were placed in cluster D and recognized lower mean ranking values. The results suggested that *Miscanthus* hybrids and *Arundo* clones might act differentially under drought and Zn stress.

## 5. Conclusions

Significant differences in response to drought and Zn tolerance among seven *Miscanthus* hybrid lines and giant reed clones, based on their growth, physiological and biochemical responses were found in climate chamber pot trials. Based on STI values of each morpho-physiological parameter, hierarchical clustering analysis (HCA) and PCA ranking value, our results showed that *Miscanthus* hybrid lines *M. sinensis* × *M. sacchariflorus* GRC10 was the most drought and Zn stress tolerant. Thereafter, *M. sinensis* × *M. sacchariflorus* line GRC15 was identified as drought tolerant, while GRC14 was Zn stress tolerant. *Arundo* clone PC1 was the most drought and Zn stress tolerant. We also found that *M. sinensis* × *M. sinensis* hybrid line GRC3 was the most sensitive hybrid for both drought and Zn stress, while CT2 was the most sensitive clone for both stresses. The physiological and biochemical measurements along with growth parameters measured in this study were effective for discerning differences between *Miscanthus* and *Arundo* and a range of variants within these species. The findings of this study provide opportunities for further reductionist experiments needed to investigate specific metabolic and their underlying molecular mechanisms providing tolerance to drought and HM stresses in these two species.

On the other hand, marginal land presents a large opportunity to produce non-food biomass from perennial grasses with low risks of Indirect Land Use Change (low ILUC). It needs to be tested if these trait-based resilience rankings will translate to biomass yield and quality performance in crops growing in real Zn contaminated and marginal soils over multiple years with weather patterns that are becoming more erratic in this era of extreme weather caused by climate change.

## Figures and Tables

**Figure 1 biology-12-01525-f001:**
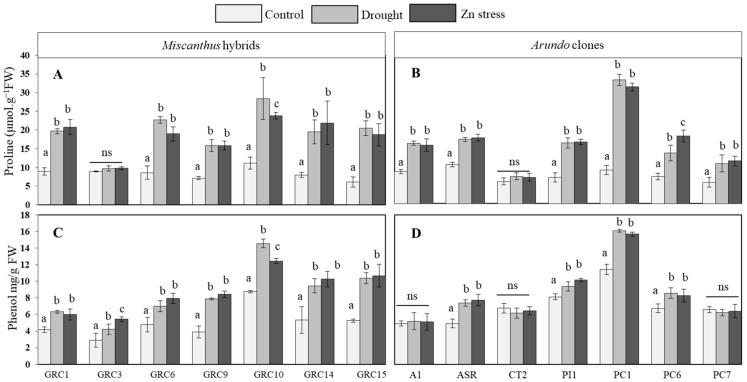
Total proline (**A**,**B**) and phenol (total phenol) (**C**,**D**) contents in leaves of *Miscanthus* hybrids and *Arundo* clones, respectively. Data are expressed as mean ± SD (*n* = 4), and different letters (a, b and c) indicate significant difference and ns = non-significant differences between control and treatments by Tukey’s HSD post hoc test at *p* < 0.05.

**Figure 2 biology-12-01525-f002:**
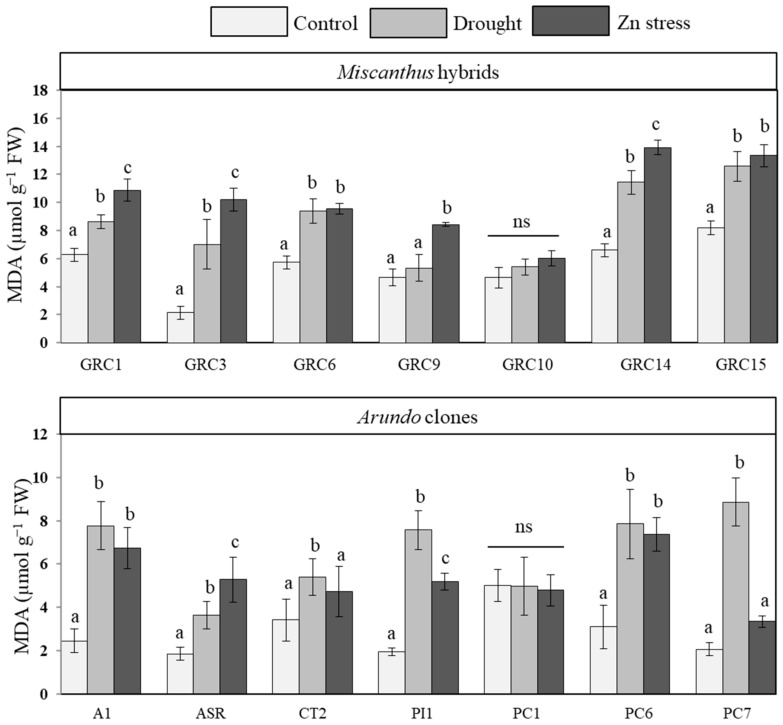
Drought and Zn both stress-induced the content of Malondialdehyde (MDA) in leaves of *Miscanthus* hybrids (**above**) and *Arundo* clones (**below**). Data are presented with mean ± SD (*n* = 4). Different letters (a, b and c) indicate significant difference and ns = non-significant differences between control and treatments by Tukey’s HSD post hoc test at *p* < 0.05.

**Figure 3 biology-12-01525-f003:**
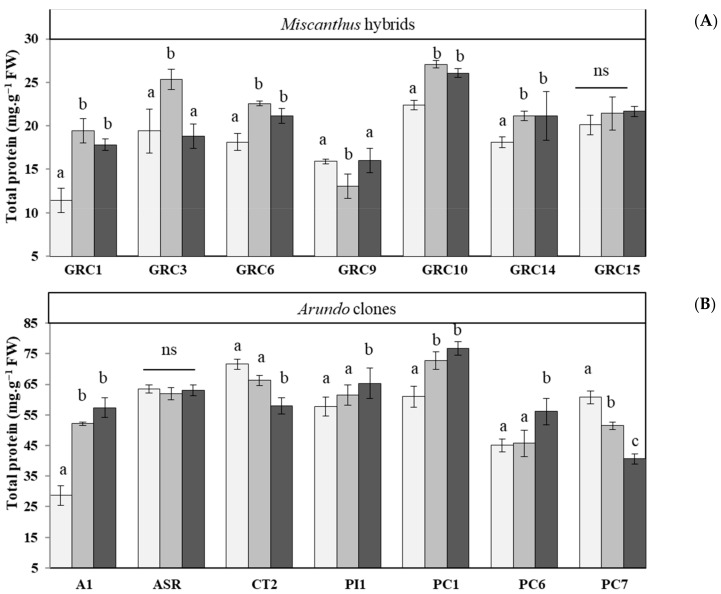
Indicate total protein content in leaves of *Miscanthus* hybrids (**A**) and *Arundo* clones (**B**), respectively. Data are presented with mean ± SD (*n* = 4). Different letters (a, b and c) indicate significant differences and ns = non-significant differences between control and treatments by Tukey’s HSD post hoc test at *p* < 0.05.

**Figure 4 biology-12-01525-f004:**
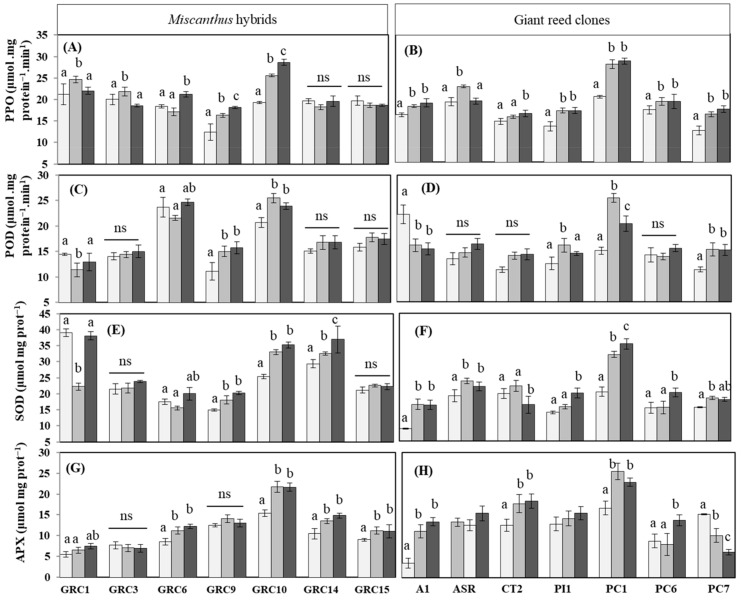
Activities of antioxidant enzymes as (**A**) Polyphenol oxidases (PPO), (**B**) peroxidase (POD), (**C**) Superoxide dismutase (SOD), (**D**) ascorbate peroxidase (APX) in the leaves of *Miscanthus* hybrids and (**E**) PPO, (**F**) POD, (**G**) SOD, (**H**) APX in *Arundo* leaves under drought and Zn stress. Data are expressed with mean ± SD (*n* = 4), and different letters (a, b and c) indicate significant differences and ns = non-significant difference between control and treatments by Tukey’s HSD post hoc test at *p* < 0.05.

**Figure 5 biology-12-01525-f005:**
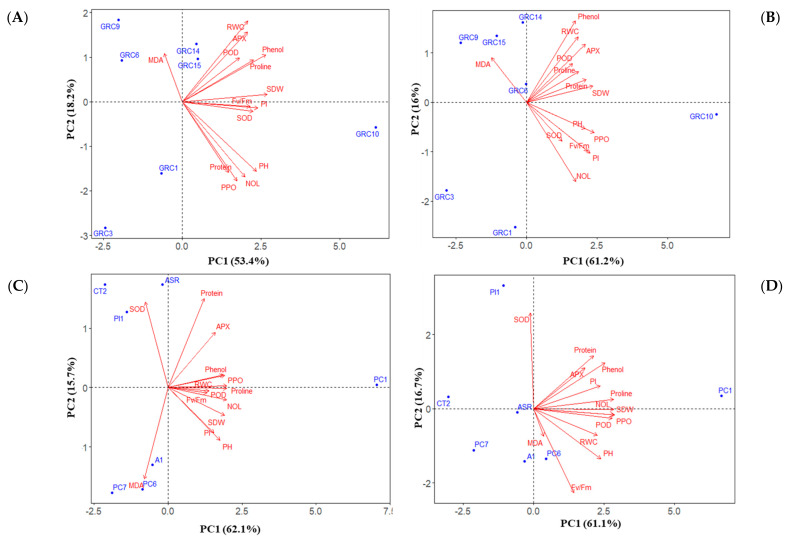
Principal components analysis (PCA) biplot of the growth and physiological parameters of seven *Miscanthus* hybrids and seven *Arundo* clones. Here PCA plots (**A**,**B**) are based on data under drought and Zn stress of *Miscanthus* hybrids and (**C**,**D**) obtained under drought and Zn stress of *Arundo* clones, respectively.

**Figure 6 biology-12-01525-f006:**
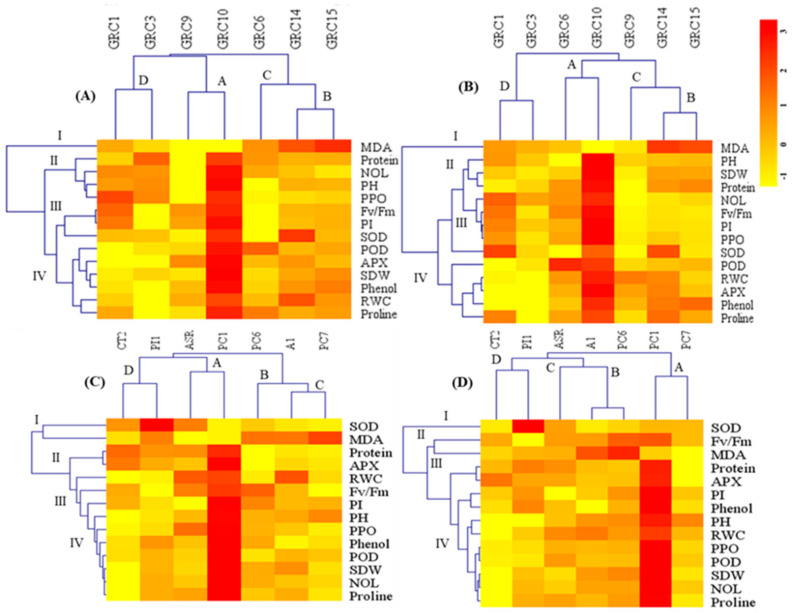
Heatmap and hierarchical clustering analysis (HCA) for growth and physiological parameters under drought and Zn stress conditions of seven *Miscanthus* hybrids and seven *Arundo* clones. HCA, (**A**,**B**) based on data under drought and Zn stress of *Miscanthus* hybrids and (**C**,**D**) obtained under drought and Zn of *Arundo* clones, respectively. The color scale is based on STI values of each trait. The clustering under drought and Zn stress for hybrids and clones was determined four groups as indicated A, B, C & D.

**Table 1 biology-12-01525-t001:** List of the seven *Miscanthus* hybrids (left), source, genotypes and seven *Arundo* clones (right) and their origin considered for this study.

*Miscanthus* Hybrids			*Arundo* Clones	
Code	Type	Hybrid	Clones Code	Origin
GRC 1	Seed-based plugs	*M. sinensis* × *M. sinensis*	A1	Italy
GRC 3	Seed-based plugs	*M. sinensis* × *M. sinensis*	ASR	Rome
GRC 6	Seed-based plugs	*M. sinensis* × *M. sinensis*	CT2	Sicily
GRC 9	Rhizomes	*Miscanthus* × *giganteus*	PI1	Tuscany
GRC 10	Seed-based plugs	*M. sinensis* × *M. sacchariflorus*	PC1	Piacenza
GRC 14	Seed-based plugs	*M. sinensis* × *M. sacchariflorus*	PC6	Piacenza
GRC 15	Rhizomes	*M. sinensis* × *M. sacchariflorus*	PC7	Piacenza

**Table 2 biology-12-01525-t002:** The growth parameters on plant height (PH) (cm), number of leaves (NOL) and shoot dry weight (SDW) (gm) of *Miscanthus* hybrids and *Arundo* clones under control, drought and Zn stress conditions. Data are presented Mean ± SD (mean ± standard deviation, *n* = 4) and different letters (a, b, and c) indicate a significant difference between control and treatments by Tukey’s post hoc test at *p* < 0.05.

	PH			NOL			SDW		
*Miscanthus* Hybrids	Control	Drought	Zn	Control	Drought	Zn	Control	Drought	Zn
GRC1	79.0 ± 7.8 a	65.6 ± 3.0 b	68.2 ± 2.0 b	9.0 ± 1.0 a	6.3 ± 0.5 b	7.0 ± 1.4 b	13.1 ± 3.9 a	6.0 ± 2.1 b	8.3 ± 0.5 b
GRC3	77.2 ± 2.2 a	67.5 ± 3.6 b	63.2 ± 2.2 b	8.5 ± 0.5 a	6.2 ± 0.5 b	6.0 ± 0.8 b	13.0 ± 3.5 a	7.5 ± 1.5 b	6.3 ± 1.2 b
GRC6	67.6 ± 1.5 a	51.3 ± 3.2 b	56.2 ± 3.4 b	8.3 ± 0.5 a	6.0 ± 0.3 b	6.2 ± 0.5 b	15.4 ± 2.3 a	7.1 ± 1.2 b	8.0 ± 0.5 b
GRC9	76.0 ± 2.6 a	51.7 ± 3.3 b	62.3 ± 1.5 c	6.0 ± 0.4 a	4.5 ± 0.5 b	4.0 ± 0.4 b	14.4 ± 1.8 a	6.6 ± 0.6 b	5.9 ± 1.8 b
GRC10	88.7 ± 4.5 a	83.5 ± 3.1 a	83.7 ± 2.6 a	8.2 ± 0.5 a	7.7 ± 0.5 a	7.7 ± 0.5 a	22.7 ± 5.3 a	20.7 ± 4.0 a	21.6 ± 2.8 a
GRC14	70.3 ± 1.5 a	63.0 ± 3.0 b	64.0 ± 1.8 b	7.3 ± 0.5 a	5.6 ± 0.5 b	5.0 ± 0.4 b	14.7 ± 2.3 a	10.3 ± 1.7 b	10.9 ± 1.7 b
GRC15	81.6 ± 2.5 a	62.2 ± 2.2 b	64.7 ± 2.2 b	6.6 ± 0.5 a	5.2 ± 0.5 b	5.2 ± 0.5 b	16.3 ± 0.3 a	11.7 ± 0.6 b	10.6 ± 1.7 b
*Arundo* clones									
A1	78.3 ± 2.8 a	57.6 ± 1.1 b	62.7 ± 2.2 c	9.3 ± 0.5 a	5.3 ± 0.5 b	6.2 ± 0.8 b	29.3 ± 0.5 a	16.1 ± 1.9 b	15.3 ± 1.6 b
ASR	70.6 ± 1.5 a	52.5 ± 1.2 b	52.5 ± 3.5 b	8.3 ± 0.5 a	5.2 ± 0.9 b	5.2 ± 0.9 b	13.6 ± 0.6 a	3.5 ± 0.3 b	3.4 ± 0.2 b
CT2	45.7 ± 4.1 a	26.2 ± 1.2 b	30.0 ± 2.1 c	6.2 ± 0.5 a	4.2 ± 0.5 b	3.5 ± 0.7 b	13.9 ± 1.1 a	1.4 ± 0.5 b	1.5 ± 0.4 b
PI1	53.2 ± 1.7 a	41.5 ± 1.2 b	36.7 ± 1.5 c	7.0 ± 0.8 a	5.7 ± 0.5 b	5.7 ± 0.5 b	19.8 ± 1.0 a	5.9 ± 0.4 b	5.5 ± 0.7 b
PC1	104.0 ± 3.6 a	97.0 ± 0.8 b	91.2 ± 0.9 c	10.7 ± 0.9 a	9.0 ± 0.3 b	10.0 ± 0.8 ab	50.9 ± 4.2 a	41.2 ± 0.7 b	42.4 ± 1.2 b
PC6	77.5 ± 1.0 a	55.2 ± 1.2 b	64.7 ± 1.5 c	8.5 ± 1.2 a	5.7 ± 0.5 b	6.2 ± 0.5 b	35.5 ± 2.7 a	13.0 ± 0.5 b	13.7 ± 0.9 b
PC7	85.3 ± 1.5 a	65.0 ± 1.0 b	66.6 ± 1.5 b	6.0 ± 1.0 a	5.0 ± 0.8 a	5.3 ± 0.6 a	20.0 ± 0.2 a	6.5 ± 0.7 b	5.7 ± 0.7 b

**Table 3 biology-12-01525-t003:** Dark-adapted chlorophyll fluorescence, the maximum quantum efficiency of the photosystem II (Fv/Fm), performance index (PI-ABS) and relative water content (RWC%) of *Miscanthus* hybrids and *Arundo* clones after 28 days of drought and Zn stress. Data are mean ± SD (*n* = 4) and values followed by different letters indicate significant differences and the same letter indicates no statistically significant difference by Tukey’s post hoc test at *p* < 0.05.

	Fv/Fm			PI-ABS			RWC%		
*Miscanthus* Hybrids	Control	Drought	Zn	Control	Drought	Zn	Control	Drought	Zn
GRC1	0.6 ± 0.03 a	0.51 ± 0.05 b	0.45 ± 0.03 c	4.25 ± 0.03 a	2.83 ± 0.39 b	2.51 ± 0.21 c	77.4 ± 2.32 a	69.0 ± 1.79 b	60.70 ± 3.89 c
GRC3	0.5 ± 0.01 a	0.26 ± 0.01 b	0.28 ± 0.01 b	3.69 ± 0.26 a	1.34 ± 0.08 b	1.74 ± 0.24 c	71.0 ± 5.40 a	52.8 ± 3.56 b	54.3 ± 4.96 b
GRC6	0.58 ± 0.00 a	0.21 ± 0.07 b	0.43 ± 0.01 c	3.52 ± 0.21 a	1.12 ± 0.38 b	2.03 ± 0.15 c	83.8 ± 3.61 a	71.4 ± 0.84 b	73.5 ± 2.33 b
GRC9	0.59 ± 0.02 a	0.43 ± 0.09 b	0.25 ± 0.01 c	3.51 ± 0.12 a	2.36 ± 0.49 b	1.31 ± 0.03 c	84.0 ± 1.47 a	73.7 ± 0.42 b	77.1 ± 0.81 ab
GRC10	0.64 ± 0.01 a	0.60 ± 0.02 a	0.60 ± 0.04 ab	4.34 ± 0.27 a	3.91 ± 0.29 b	3.90 ± 0.45 b	97.2 ± 7.04 a	87.2 ± 6.06 b	88.6 ± 6.88 b
GRC14	0.51 ± 0.02 a	0.38 ± 0.01 b	0.28 ± 0.04 c	3.68 ± 0.16 a	2.05 ± 0.12 b	1.52 ± 0.22 c	94.6 ± 4.91 a	85.9 ± 3.54 b	75.5 ± 3.44 c
GRC15	0.53 ± 0.03 a	0.38 ± 0.01 b	0.27 ± 0.03 c	3.74 ± 0.28 a	2.19 ± 0.17 b	1.46 ± 0.17 c	83.9 ± 3.35 a	76.8 ± 1.96 a	65.1 ± 2.94 b
*Arundo* clones									
A1	0.81 ± 0.02 a	0.65 ± 0.07 b	0.70 ± 0.01 b	5.26 ± 1.67 a	2.94 ± 1.28 b	2.44 ± 0.50 b	90.7 ± 0.08 a	75.9 ± 0.33 b	79.0 ± 0.69 c
ASR	0.78 ± 0.07 a	0.73 ± 0.04 a	0.70 ± 0.06 a	2.62 ± 1.03 a	0.74 ± 0.72 b	0.57 ± 0.35 b	92.9 ± 0.24 a	75.4 ± 0.78 b	77.3 ± 0.85 b
CT2	0.80 ± 0.03 a	0.67 ± 0.06 b	0.68 ± 0.05 b	4.22 ± 2.25 a	3.16 ± 1.49 a	2.46 ± 1.20 ab	88.7 ± 0.41 a	64.1 ± 0.64 b	58.8 ± 1.88 c
PI1	0.81 ± 0.03 a	0.50 ± 0.10 b	0.40 ± 0.25 b	4.78 ± 1.57 a	2.13 ± 0.72 b	3.73 ± 2.04 ab	93.1 ± 0.97 a	65.5 ± 0.53 b	71.2 ± 0.26 c
PC1	0.82 ± 0.01 a	0.82 ± 0.02 a	0.81 ± 0.01 a	7.25 ± 0.91 a	6.78 ± 0.32 a	6.98 ± 2.31 a	92.4 ± 1.93 a	76.8 ± 1.61 b	84.5 ± 1.81 c
PC6	0.81 ± 0.03 a	0.78 ± 0.05 a	0.80 ± 0.02 a	4.93 ± 0.83 a	4.10 ± 2.17 a	3.74 ± 1.47 a	90.2 ± 0.40 a	65.9 ± 1.89 b	75.8 ± 1.19 c
PC7	0.81 ± 0.01 a	0.56 ± 0.08 b	0.66 ± 0.08 b	4.52 ± 1.24 a	3.14 ± 0.86 a	2.57 ± 1.41 a	87.0 ± 1.70 a	67.0 ± 1.55 b	73.1 ± 1.83 c

**Table 4 biology-12-01525-t004:** Summary of analysis of variance (two-factor ANOVA) for the effects of treatments (drought and Zn), hybrids of *Miscanthus* (top) and clones of *Arundo* (below) and the interactions on plant height (PH), number of leaves (NOL), shoot dry weight (SDW), the maximum quantum efficiency of the PSII (Fv/Fm), performance index (PI-ABS), leaf relative water content (RWC), protein, enzymes including polyphenol oxidase (PPO), peroxidase (POD), superoxide dismutase (SOD) and ascorbate peroxidase (APX), lipid peroxidation (MDA), phenol and proline with data after 28 days of stress.

Variable	PH	NOL	SDW	Fv/Fm	PI-ABS	RWC%	Protein	PPO	POD	SOD	APX	MDA	Phenol	Proline
*Miscanthus* hybrids														
Treatments	***	***	***	***	***	***	***	***	***	***	***	***	***	***
Hybrids	***	***	***	***	***	***	***	***	***	***	***	***	***	***
Treatments × hybrids	***	*	NS	***	***	**	***	***	***	***	***	***	***	***
*Arundo* clones														
Treatments	***	***	***	***	***	***	***	***	***	***	***	***	***	***
Clones	***	***	***	***	***	***	***	***	***	***	***	***	***	***
Treatments × clones	***	**	***	***	NS	***	***	***	***	***	***	***	***	***

** Significant at *p* ≤ 0.01, *** significant at *p* ≤ 0.001, * significant at *p* < 0.05, NS nonsignificant at *p* > 0.05.

**Table 5 biology-12-01525-t005:** PCA ranking values are based on stress tolerance index (STI) with three major principal components (PC1, PC2, and PC3) and numeric ranking of *Miscanthus* hybrids and *Arundo* clones under drought and Zn stress.

*Miscanthus*											
Drought						Zn					
Hybrids	PC1	PC2	PC3	Ranking	Numeric Rank	Hybrids	PC1	PC2	PC3	Ranking	Numeric Rank
GRC10	15.54	−0.85	−0.28	8.11	1	GRC10	18.24	−0.33	−0.62	11.34	1
GRC15	1.25	1.42	1.56	2.14	2	GRC14	−0.33	2.24	2.26	1.32	2
GRC14	1.13	1.91	0.44	1.96	3	GRC6	−0.02	0.51	−1.92	−0.16	3
GRC1	−1.69	−2.38	−1.92	−1.58	4	GRC1	−1.05	−3.50	1.90	−0.95	4
GRC6	−4.87	1.37	2.09	−2.09	5	GRC15	−2.86	1.87	0.66	−1.45	5
GRC9	−5.15	2.71	−2.70	−2.60	6	GRC9	−6.30	1.67	−1.09	−3.84	6
GRC3	−6.21	−4.18	0.82	−3.97	7	GRC3	−7.69	−2.47	−1.19	−5.32	7
** *Arundo* **											
**Drought**						**Zn**					
**Clones**	**PC1**	**PC2**	**PC3**	**Ranking**	**Numeric Rank**	**Clones**	**PC1**	**PC2**	**PC3**	**Ranking**	**Numeric Rank**
PC1	19.30	0.06	0.51	12.22	1	PC1	18.04	0.48	−0.12	11.39	1
ASR	0.99	6.18	9.41	2.45	2	PC6	1.22	−1.92	0.79	1.56	2
A1	−1.53	−1.86	−0.89	−1.33	3	A1	−0.79	−2.04	0.25	−0.79	3
PC6	−2.44	−2.41	−0.30	−1.93	4	ASR	−1.60	−0.17	0.22	−1.01	4
PI1	−3.80	1.64	2.18	−1.94	5	PI1	−3.16	4.65	−0.80	−1.33	5
PC7	−5.19	−2.34	0.64	−3.57	6	PC7	−6.14	−1.64	−2.49	−4.34	6
CT2	−5.84	2.37	0.20	−3.31	7	CT2	−9.76	0.57	2.13	−5.84	7

## Data Availability

The authors confirm that the data supporting the findings of this study are available within the article [and/or] its Appendix A.

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
