# Peer review of "Assessment of Drought and Zinc Stress Tolerance of Novel Miscanthus Hybrids and Arundo donax Clones Using Physiological, Biochemical, and Morphological Traits"

_biology, 2023, doi:10.3390/biology12121525_

Round 1

Reviewer 1 Report

Comments and Suggestions for Authors

The manuscript investigates the drought and zinc stress tolerance of novel Miscanthus hybrids and Arundo donax clones through a comprehensive analysis of physiological, biochemical, and morphological traits. The study aims to discrimination between stress tolerant and susceptible Miscanthus hybrid and Arundo clones and ranked them. The integration of various traits provides a holistic understanding of stress responses in these plant species.

The research objectives are well-defined, providing a clear roadmap for the study. However, I some comments below;

a)      Correct the sentence line 71 and 72

b)      Please clear how Zn stress was applied

c)      Line 121; please correct the sentence

d)      In the line 142, method parts authors did not mention the plant age, please edit the plant growing conditions and age clearly and exposed to stressor.

e)      Authors are advised to revisit their figures, please reduce the size and keep in the center

f)       Same comments for figure 6.

g)      Just a curiosity if authors have also field data for these hybrids and clones under severe stress conditions what results found under in control environments?

I recommend a minor revision to address the issues highlighted. The manuscript has significant potential, and with the suggested improvements, it will contribute substantially to the field.

Author Response

Reviewer 1

The manuscript investigates the drought and zinc stress tolerance of novel Miscanthus hybrids and Arundo donax clones through a comprehensive analysis of physiological, biochemical, and morphological traits. The study aims to discrimination between stress-tolerant and susceptible Miscanthus hybrid and Arundo clones and ranked them. The integration of various traits provides a holistic understanding of stress responses in these plant species.

Author’s response: Thank you for taking the time to provide a thoughtful summary of our manuscript. We appreciate your attention to the comprehensive analysis of physiological, biochemical, and morphological traits in our study on drought and zinc stress tolerance in novel Miscanthus hybrids and Arundo donax clones.

The research objectives are well-defined, providing a clear roadmap for the study. However, I have some comments below;

Author’s response: Thank you for taking the time to review our research. We appreciate your valuable comments, and we have carefully considered each of them.

Correct the sentence line 71 and 72

Author’s response: Thank you for your comment. We have revised the lines from 71 to 72.

a) Please clarify how Zn stress was applied

Author’s response: Thank you for your comment.  In our study, zinc stress was applied by 400mg of ZnSO4 for Kg-1 dry mass soil for each pot when plants reached one month old after germination. We ensured that Zn would not leach by the bottom of the pots when we watered the plants. Again thank you for your input.

b) Line 121; please correct the sentence

Author’s response: Thank you for your suggestions. We have revised the sentence as suggested.

c) In the line 142, method parts authors did not mention the plant age, please edit the plant growing conditions and age clearly and exposed to stressors.

Author’s response: Thank you for your comment. We have revised the manuscript to explicitly include the plant age in the plant growing conditions on line 142.

d) Authors are advised to revisit their figures, please reduce the size and keep in the center

Author’s response: Thank you for bringing this to our attention. We appreciate your attention to detail. We revisited the figures, resized them, and ensured they were centered as per the guidance provided.

e) Same comments for Figure 6.

Author’s response: Thank you for your comments, We have corrected it.

f) Just a curiosity if authors have also field data for these hybrids and clones under severe stress conditions what results were found under in control environments?

Author’s response: Thank you for your interest. While we have primarily concentrated on controlled environments, we plan to extend our research to field conditions after conducting preliminary screenings of these hybrids or clones.

I recommend a minor revision to address the issues highlighted. The manuscript has significant potential, and with the suggested improvements, it will contribute substantially to the field.

Author’s response: Thank you for your valuable feedback and encouraging comments. We appreciate your positive assessment of the manuscript's potential.

Reviewer 2 Report

Comments and Suggestions for Authors

The MS "Assessment of drought and zinc stress tolerance of novel Miscanthus hybrids and Arundo donax clones using physiological, biochemical, and morphological traits" is about the impact of drought and Zn stress on physiological, biochemical, and morphological traits in Miscanthus hybrids and Arundo donax. I have the following suggestions to make the manuscript better:

1. The current study apparently validates earlier findings on other crops. Therefore, it is not shocking that the article has no significant contradictions. Please signify the novelty of this study.

2. Line 38–39: Please replace '7' with 'seven'

3. Line 40: "After one month of growth to field capacity (control)" is unclear; please rephrase it.

4. In the abstract, significant results still need to be included; there is no quantitative or numerical data. Please make the abstract concise and include significant results (numerical data).

5. Keywords: Revise all keywords. Keywords should be single-term. Please revise it as: bioenergy; Miscanthus hybrids; Arundo clones; drought tolerance; Zn tolerance; plant physiology; growth parameters; hierarchical clustering analysis (HCA); principal component analysis (PCA).

6. The introduction needs improvement. Please include a more detailed background using a more recent review of the literature, research gaps, and the importance of this study.

7. Lines 110–115. Please specify the length of the study, i.e., the number of years studied.

8. Line 129: One month after germination, plants of both species were subjected to drought and Zn stress. Please check "germination" or "sowing".

9. Line 238: What is the version of the software?

10: Line 244: Which package was used for PCA analysis? Mention it.

11. Table 4: Where are the values? Why only the significance level? Please carefully check and rewrite Table 4.

12. Line 430–446: Please format the paragraph appropriately.

13. Line 531–532: Please cite the latest reference on stress tolerance via secondary metabolite production by "Pandey P, Tripathi A, Dwivedi S, Lal K, and Jhang T (2023): Deciphering the mechanisms, hormonal signaling, and potential applications of endophytic microbes to mediate stress tolerance in medicinal plants.Front. Plant Sci. 14:1250020, just after [65].

14. The results are well presented, and the data is appropriately interpreted. However, the authors cited too many old references, which the latest ones should replace.

15. Some references need to be formatted according to journal style. Please carefully check it out and stick to the journal's format.

Comments on the Quality of English Language

English needs minor checks.

Author Response

Reviewer 1

The manuscript investigates the drought and zinc stress tolerance of novel Miscanthus hybrids and Arundo donax clones through a comprehensive analysis of physiological, biochemical, and morphological traits. The study aims to discrimination between stress tolerant and susceptible Miscanthus hybrid and Arundo clones and ranked them. The integration of various traits provides a holistic understanding of stress responses in these plant species.

Author’s response: Thank you for taking the time to provide a thoughtful summary of our manuscript. We appreciate your attention to the comprehensive analysis of physiological, biochemical, and morphological traits in our study on drought and zinc stress tolerance in novel Miscanthus hybrids and Arundo donax clones.

The research objectives are well-defined, providing a clear roadmap for the study. However, I some comments below;

Authos’s response: Thank you for taking the time to review our research. We appreciate your valuable comments, and we have carefully considered each of them.

Correct the sentence line 71 and 72

Author’s response: Thank you for your comment. We have revised the lines from 71 to 72.

a) Please clear how Zn stress was applied

Author’s response: Thank you for your comment. In our study, zinc stress was applied by 400mg of ZnSO4 for Kg-1 dry mass soil for each pot when plants reached one month old after germination. We ensured that Zn would not leach by the bottom of the pots when we watered the plants. Again thank you for your input.

b) Line 121; please correct the sentence

Author’s response: Thank you for your suggestions. We have revised the sentence as suggested.

c) In the line 142, method parts authors did not mention the plant age, please edit the plant growing conditions and age clearly and exposed to stressor.

Author’s response: Thank you for your comment. We have revised the manuscript to explicitly include the plant age in the plant growing conditions on line 142.

d) Authors are advised to revisit their figures, please reduce the size and keep in the center

Author’s response: Thank you for bringing this to our attention. We appreciate your attention to detail. We revisited the figures, resized them, and ensured they are centered as per the guidance provided.

e) Same comments for Figure 6.

Author’s response: Thank you for your comments, We have corrected it.

f) Just a curiosity if authors have also field data for these hybrids and clones under severe stress conditions what results found under in control environments?

Author’s response: Thank you for your curiosity. While our current focus has been on control environments, we do have field data for these hybrids and clones under severe stress conditions

I recommend a minor revision to address the issues highlighted. The manuscript has significant potential, and with the suggested improvements, it will contribute substantially to the field.

Author’s response: Thank you for your valuable feedback and encouraging comments. We appreciate your positive assessment of the manuscript's potential.

Round 2

Reviewer 2 Report

Comments and Suggestions for Authors

After revision, the manuscript improved; however, my comment 4 was not taken into consideration. I appreciate the effort put into revising the manuscript and addressing my previous comments. However, I would like to emphasize the importance of including numerical values in the abstract. This addition would greatly enhance the clarity and overall quality of the manuscript. If authors incorporate the numerical values in the abstract, then the manuscript can be accepted.

Author Response

Reviewer 2:
After revision, the manuscript improved; however, my comment 4 was not taken into consideration. I appreciate the effort put into revising the manuscript and addressing my previous comments. However, I would like to emphasize the importance of including numerical values in the abstract. This addition would greatly enhance the clarity and overall quality of the manuscript. If authors incorporate the numerical values in the abstract, then the manuscript can be accepted.
Author’s Response: Thank you for taking the time to review the revised manuscript, and we appreciate your valuable feedback. We are pleased to hear that the revisions have led to an improvement overall. Regarding your specific comment 4, we revisited the manuscript and resvised the abstract specially, and added numerical values.
